# Retrospective National “Real Life” Experience of the SFCE with the Metronomic MEMMAT and MEMMAT-like Protocol

**DOI:** 10.3390/jcm12041415

**Published:** 2023-02-10

**Authors:** Camille Winnicki, Pierre Leblond, Franck Bourdeaut, Anne Pagnier, Gilles Paluenzela, Pascal Chastagner, Gwenaelle Duhil-De Benaze, Victoria Min, Hélène Sudour-Bonnange, Catherine Piette, Natacha Entz-Werle, Sylvie Chabaud, Nicolas André

**Affiliations:** 1Department of Pediatric Immunology, Hematology and Oncology, Children Hospital of La Timone, Assistance Publique Hôpitaux de Marseille, 13005 Marseille, France; 2Department of Pediatric Oncology, Institut d’Hématologie et d’Oncologie Pédiatrique, Centre Léon Bérard, 69008 Lyon, France; 3SIREDO Pediatric Oncology Center, Curie Institute, 75005 Paris, France; 4Department of Pediatric Immunohematology and Oncology, University Hospital, 38043 Grenoble, France; 5Department of Pediatric Hematology-Oncology, Centre Hospitalo-Universitaire de Montpellier, 34000 Montpellier, France; 6Pediatric Oncology, University Hospital of Nancy, 54000 Nancy, France; 7Department of Pediatric Oncology, Centre Hospitalier Universitaire, University Côte d’Azur, 06108 Nice, France; 8Oscar-Lambret Center, Department of Pediatric Oncology & AYA Unit, 59020 Lille, France; 9Department of Pediatric Oncology, Centre Hospitalo-Universitaire de Liège, 4000 Liège, Belgium; 10Pediatric Onco-Hematology Department-Pediatrics III, University Hospital of Strasbourg, 67091 Strasbourg, France; 11Department of Statistics, Centre Léon Bérard, 69373 Lyon, France; 12Centre de Recherche en Cancérologie de Marseille, Aix-Marseille Université, Inserm, CNRS, 13273 Marseille, France; 13Metronomics Global Health Initiative, 13385 Marseille, France

**Keywords:** pediatric oncology, brain tumors, metronomic chemotherapy, pharmacology, medulloblastoma, ATRT, angiogenesis, immunotherapy

## Abstract

Background: Relapses in pediatric high-risk brain tumors remain unmet medical needs. Over the last 15 years, metronomic chemotherapy has gradually emerged as an alternative therapeutic approach. Patients and Methods: This is a national retrospective study of patients with relapsing pediatric brain tumors treated according to the MEMMAT or MEMMAT-like regimen from 2010 to 2022. Treatment consisted of daily oral thalidomide, fenofibrate, and celecoxib, and alternating 21-day cycles of metronomic etoposide and cyclophosphamide associated with bevacizumab and intraventricular chemotherapy. Results: Forty-one patients were included. The most frequent malignancies were medulloblastoma (22) and ATRT (8). Overall, the best responses were CR in eight patients (20%), PR in three patients (7%), and SD in three patients (7%), for a clinical benefit rate of 34%. The median overall survival was 26 months (IC95% = 12.4–42.7), and median EFS was 9.7 months (IC95% = 6.0–18.6). The most frequent grade ¾ toxicities were hematological. Dose had to be adjusted in 27% of the cases. There was no statistical difference in outcome between full or modified MEMMAT. The best setting seems to be when MEMMAT is used as a maintenance and at first relapse. Conclusions: The metronomic MEMMAT combination can lead to sustained control of relapsed high-risk pediatric brain tumors.

## 1. Introduction

Central nervous system (CNS) cancers are the most frequent group of solid tumors in children and represent almost 3000 patients per year in Europe [1,2,3,4]. Initial treatments usually include primary surgery, radiation therapy, and chemotherapy [1,2,5,6]. Targeted molecules are gaining growing interest in some specific situations such as BRAF pathway-altered glioma [7,8] or NTRK-altered brain tumors [9,10]. Nevertheless, overall survival remains poor in some tumors, and this is even more critical for relapsed or resistant malignancies such as high-grade gliomas or medulloblastoma. Consequently, new therapeutic strategies are needed. One such potential therapeutic approach is metronomic chemotherapy (MC). MC relies on the frequent administration of a low dose of chemotherapy without long breaks [11,12]. It is frequently combined with drug repurposing to generate metronomics. Its mechanism of action is complex and based on multi-target effects. Indeed, MC was initially reported to be anti-angiogenic, but its pro-immune anti-tumoral effect are being increasingly described [13,14,15]. Additionally, direct effects on anti-cancer and anti-cancer stems cells seem to contribute to the activity of MC [16].

In children, several reports have indicated the potential of various metronomic combinations to control relapsing/refractory tumors [11,17,18,19,20], and its interest during maintenance is well-known in leukemia and growing in solid tumors such as rhabdomyosarcoma [21]. Of note, the use of MC also seems to be a promising and well-suited strategy for low- and middle-income countries [19,22,23]. 

Among many metronomic regimens, the four-drug regimen initially reported by Kieran et al. in a phase 2 trial showed a good safety profile as well as sustained tumor control, especially in ependymoma [17]. This combination was then completed with the addition of fenofibrate, which did not seem to significantly improve its efficacy, as reported in a larger phase 2 trial [24]. The combination was then further enriched by the addition of intra-ventricular (IVe) chemotherapy (etoposide and aracytine) and bevacizumab to generate the so-called MEMMAT combination for metronomic multitarget anti-angiogenic therapy [25,26]. Intra-ventricular chemotherapy was introduced to target meningeal disease that does not seem to be angiogenic-dependent. Bevacizumab was added to further strengthen the anti-angiogenic effect of the combination, as previously reported in the initial metronomic preclinical publications [27]. The initial pilot study reported promising results especially in medulloblastoma [26]. An international state-of-the-art phase II study is ongoing (NCT01356290).

In France, several centers have started treating pediatric patients with refractory/relapsing brain tumors according to the MEMMAT regimen outside of the trial. Interestingly, physicians also sometimes used lighter versions of this protocol, for instance, by not using intra-ventricular chemotherapy, bevacizumab, or thalidomide. We report here the retrospective experience of the French Society for Children with Cancer (SFCE).

## 2. Materials and Methods

### 2.1. Patients and Data Collection

In this retrospective study, data from patients younger than 19 years treated in pediatric oncology units of the SCFE from 2010 to 2022 who received the MEMMAT combination were collected. All 41 patients had relapsed or refractory CNS brain tumors. A full medical history was obtained from the electronic medical chart including neurological examination; performance status evaluation; and routine laboratory tests including blood chemistry, urine analysis, CSF analysis, and MRI scans. MRI scans were repeated at least every 2–3 months during treatment and during follow-up until progression.

### 2.2. The MEMMAT Regimen

Treatment consisted of a continuous oral regimen including the following:-Daily oral thalidomide (3 mg/kg/d);-Daily oral fenofibrate (90 mg/m^2^/d);-Twice daily oral celecoxib (100 to 400 mg 2x/d);-Alternating 21-day cycles of low-dose oral etoposide (50 mg/m^2^/d) and cyclophosphamide (2.5 mg/kg/d) [24].-Intra-venous Bevacizumab (10 mg/kg) every two weeks;-IVe therapy consisting of alternating etoposide (0.25 mg to 0.5 mg/d) for five consecutive days [28], alternating with liposomal cytarabine (25–50 mg) in combination with oral steroids to prevent chemical meningitis every 3 weeks [29]. When liposomal cytarabine was no longer available, it was switched for standard aqueous aracytine.

The MEMMAT-like regimen was defined as MEMMAT without IVe therapy, without bevacizumab, and/or without thalidomide. The planned treatment duration was 1 year. In cases of toxicity, dose reductions were recommended to try to avoid treatment breaks at the discretion of the local treating physician. Additional radiotherapy concomitant to or at the end of MEMMAT was allowed in case of a response.

### 2.3. Evaluation

#### 2.3.1. Response to Treatment

T1- or T2-weighted images of the target lesions on MRI were used to evaluate the length of the 2 longest tumor dimensions. Complete response (CR) was defined as the complete disappearance of a measurable disease, a partial response (PR) was defined as ≥50% decrease in the product of the two maximum perpendicular diameters compared with the baseline evaluation, and stable disease was defined as ≤50% decrease and ≤25% increase in the product of diameters. Progressive disease (PD) was defined as a ≥25% increase in the product of diameters or the appearance of new lesions (RAPNO) [30].

#### 2.3.2. Evaluation of Toxicity

Side effects were retrospectively collected and evaluated according to the Common Terminology Criteria for Adverse Events (CTCAE) v. 4.0.

### 2.4. Statistical Methods

Progression-free survival (PFS) was defined as time elapsed from recurrence that triggered the initiation of the “MEMMAT” regimen to the date of relapse, progression, or death from any cause or, for patients without any events, to the date of last follow-up. Overall survival (OS) was defined as the time elapsed from the date of relapse that triggered the initiation of the “MEMMAT” regimen to the date of death from any cause or, for survivors, to the date of last follow-up.

PFS and OS were estimated using the Kaplan–Meier method and described in terms of median along with the associated 2-sided 95% CIs. Survival distributions were compared according to the type of malignancy; the number of previous lines of treatments; and the type of regime, full or MEMMAT-like, using a Log-Rank test, supported by a Cox regression. The hazard ratios between subgroups, along with the associated 2-sided 95% CIs for the estimates, were determined. Statistical analysis was conducted using SAS statistical software version 9.4.

## 3. Results

Forty-one patients were identified and included in this series. The characteristics of the patients are detailed in Table 1. All 41 patients had relapsed or refractory CNS brain tumors, mostly medulloblastoma (22 pts—54%) or ATRT (8 pts—27%). Thirty-three patients (80%) received MEMMAT for a relapsed/progressing tumor, and eight patients (20%) received treatment as part of maintenance. The MEMMAT treatment was given as a second-line treatment in 32% of the patients and as third- to fifth-line treatments in 68% of the patients. Almost all patients had anterior chemotherapy (95%—39 pts) and/or radiotherapy (95%—39 pts) and/or surgery (98%—40 patients). The full MEMMAT was given in 39% of the patients. Fourteen and ten patients did not receive IVe therapy and bevacizumab, respectively. Of note, seven patients also received radiotherapy during or after completing their MEMMAT treatment.

The median duration of treatment was 32 weeks (range 4–156), and the mean time to response was 75 days (+/− 97). Treatment was overall well tolerated. Grade 3 or 4 toxicities were reported in 68% of the patients. The most frequent grade 3–4 toxicities were hematological. Details of the grade 3–4 are presented in Table 2. Dose had to be adjusted in 27% of the cases.

With a mean follow-up of 28 months, the best responses were CR in eight patients (20%), PR in three patients (7%), and SD in three patients (7%), for a clinical benefit rate of 34%. Progressive disease was observed in 56% of the cases and was either local (20%) or metastatic (46%). Twenty-one patients died of disease during the follow-up period. The median overall survival was 26 months (IC95% = 12.4–42.7), and median EFS was 9.7 months (IC95% = 6.0–18.6) (Figure 1).

When considering the underlying malignancy, EFS for ATRT and medulloblastoma were, respectively, 6 months and 9.7 months. Interestingly, for the five ependymoma patients, the median EFS was not reached and only one event was reported (Figure 2). Among the seven patients who received radiotherapy during or at the end of the MEMMAT regimen, we observed two complete remissions, one partial remission, one stable disease, and three progressive diseases.

Since a significant proportion of the patients did not receive the full MEMMAT regimen, we investigated whether this had a significant impact on survival. As shown in Figure 3, there was no impact on event-free survival when comparing full versus adapted MEMMAT (7 months (4.1-NE)) vs. 13 months ((3.0–35) *p* = 0.9)). Interestingly, when looking at further details of the impact of IVe therapy or bevacuzimab on EFS, we did not find any difference either (bevacizumab (7.4 months (1.4-NE) vs. 11.5 months (6.0–18.6) *p* = 0.68) or IVe therapy (16.6 (3–35) vs. 8 months (3.9-NE) *p* = 0.73)). Lastly, a similar specific analysis run on the medulloblastoma population did not show any difference for PFS either (17 months (1–26) vs. 20 (4.5—not reached) *p* = 0.44)).

To try to identify patients more likely to benefit from the MEMMAT regimen, we investigated the impact of the number of previous lines of treatment and the setting of initiation of MEMMAT (maintenance vs. progressive disease). As shown in Figure 4, the number of previous lines of treatment was associated with a statistically significant impact on EFS. Indeed, when patients received MEMMAT as a second line of treatment, EFS was not reached vs. 6.4 months (5.7–12.9; *p* = 0.0076) for patients receiving MEMMAT as a third or higher line of treatment. Similarly, when MEMMAT was initiated as part of maintenance, EFS was significantly better in the maintenance arm (median EFS was not reached versus 7.8 months (4–17) when compared with patients with progressive disease at initiation of MEMMAT). These results suggest that MEMMAT should be used as a means of maintenance during the first relapse. Among the patients who received MEMMAT as part of maintenance, three had medulloblastoma, two had ependymoma, and three patients had other types of tumors. Out of the eight patients, four had metastatic relapse. All patients but one previously received a combination of surgery, radiotherapy, and chemotherapy. One patient out of eight was in complete remission at the initiation of MEMMAT. At last follow-up, one patient presented a progressive disease; three and two patients were in complete remission and partial remission, respectively; and two patients had a stable disease.

Lastly, as limited data are available about the use of MEMMAT of ependymoma, we then focused on the patients with ependymoma. The details are provided in Table 3. All patients had progressive disease after at least one line of treatment and previously received at least radiotherapy, except one patient who received MEMMAT as part of maintenance. Only one patient with metastatic disease progressed while on MEMMAT given as a fourth line of treatment.

## 4. Discussion

We report here the experience of the SFCE with the “real life” use of the MEMMAT or MEMMAT-like regimen outside of a clinical trial in 41 pediatric patients with refractory or relapsing brain tumors. The combination appears to be safe and to display efficacy in patients with ependymoma, ATRT, and medulloblastoma. Indeed, 25% of the patients were alive without a progressive disease after 3 years.

Among the 17 patients with medulloblastoma, we observed 10% overall survival at 24 months after initiating the MEMMAT treatment. This does not seem to be as good as the initial report of the MEMMAT regimen [26]. The Vienna team has also further reported that, out of 29 patients with medulloblastoma treatment with MEMMAT, 9 patients were alive, with a median of 44 months after recurrence [31]. Five out of nine surviving patients are currently in CR between 96 and 164 months after starting MEMMAT therapy. Of note, in this series, five patients died of another cause (accident, leukemia, and septicemia). OS was 44 ± 10% at 5 years and 39 ± 10% at 10 years, and PFS was 33 ± 10% at 5 years and 28 ± 9% at 10 years. It is not clear why the results we report here are not as good. First, at last follow-up, five patients were still under treatment in hopes that, with a longer follow-up, overall outcome might improve. Additionally, the main difference from the Vienna experience is that a significant proportion of the patients we report here did not receive the full MEMMAT regimen. Anyhow, when looking more closely at the impact of not receiving the full MEMMAT regimen, we could not find any difference in survival both in the global population or in the medulloblastoma sub-population. These findings shall nevertheless be considered cautiously because the reasons for not giving full MEMMAT (i.e., no meningeal disease and frail patients) might have induced a strong bias and could have not led to us trying to use the MEMMAT-like regime until our findings were confirmed. This study is also retrospective and was not well equipped to demonstrate the differences between the full and adapted MEMMAT. Anyhow, as the statistical analysis did not reveal significant differences in the patients receiving the full MEMMAT or MEMMAT-like regimen (i.e., without IVe therapy, thalidomide, or bevacizumab), the fact that some patients can reach sustained CR without the full regimen raises the question of an optimal design of the combination. Indeed, the results we report here seem to be anyhow better than the previous versions of the MEMMAT backbone protocol relying on a four-drug combination or five-drug combination [17,24]. The mechanism of action of MEMMAT is rooted in the metronomic concept; it was therefore designed as an antiangiogenic treatment [26,32], and IVe therapy was added to avoid resistance in the metastatic meningeal disease, which is not sensitive to MC. Furthermore, MC has been demonstrated to restore the anti-cancer properties of the immune system and to directly target cancer cells and cancer stem cells [20]. Ultimately, the MEMMAT regimen may be regarded as a treatment that targets cancer as a system, suggesting that the treatment might work differently according to the disease, patients, and setting (i.e., bulky disease vs. minimum residual disease) so that all agents may not be mandatory for all patients. Noteworthy from this perspective is that survival is better when MEMMAT is given as maintenance of maintenance after additional treatment at relapse (i.e., surgery, re-irradiation, and chemotherapy), although several treatments make it impossible to formally demonstrate the respective impacts of the different parts of the treatment on the outcome.

We also report an interesting outcome in patients with relapsed ATRT or ependymoma. The two groups of tumors represent tumors with unmet needs, and innovative therapies such as targeted therapies and immunotherapies have had quite a limited impact so far. While the number of patients remains small and shall be confirmed, MC has previously been reported to be of interest in these two types of tumors [33,34,35,36]. For ependymoma, the articulation of a metronomic MEMMAT method of maintenance may be of high value after second surgery and/or re-irradiation.

A significant proportion of the patients presented with grade 3 or grade 4 toxicities. The most frequent types were hematological toxicities that required dose adjustments. This is in part related to previous heavy treatment with craniospinal radiotherapy and/or high-dose chemotherapy followed by peripheral stem cell transplantation, but this treatment, although metronomic, has an intrinsic toxicity, as previously reported by Peyrl et al. [26] or Porkholm et al. in a modified version of the MEMMAT protocol for patients with relapsing DIPG/HGG [37]. Pediatric patients living in low- and middle-income countries (LMIC) are potentially interested in MC given its low cost and low toxicity [22]. Anyhow, the addition of bevacizumab or thalidomide, and the requirements of IT therapy make it not fully adapted for this setting. Whether a lighter MEMMAT-like protocol would be of interest for LMIC remains an open question.

We also need to acknowledge the limitations of this series. It is retrospective, despite a detailed and precise protocol; as mentioned above, the treatment is heterogenous, and several types of disease have been included. Additionally, physicians may have chosen to add IVe therapy when meningeal disease was evidenced, creating bias. Lastly, we did not have precise molecular subgroups for most patients. Group 4 medulloblastoma, for instance, has been reported to have a less aggressive behavior [38]. The repartition of the molecular subtypes of brain tumors included might therefore influence the EFS in this series. Anyhow, in the series from Slavc et al. [31] among the 23 patients whose tumors underwent molecular subgrouping classification using a DNA methylation array and brain classifier mnp-V12.5, group 4 medulloblastomas seem to group 4 have better outcome than group 3 medulloblastoma (median survival: 43 months (14–71) vs. not reached). As group 3 medulloblastomas are commonly accepted as displaying worse prognoses, the findings reported by Slavc et al. [31] seem to further highlight the potential of MEMMAT for both relapsing Group 3 and 4 medulloblastomas, which represent medical unmet needs. Moreover, the lack of molecular profiling for most of our patients should not diminish the interesting outcomes reported despite not furthering research in precision medicine. Nevertheless, this series represents real-life data from patients who have not been highly selected as it would be the case for an early phase clinical trial, and it provides valuable additional information regarding the potential of this combination.

## 5. Conclusions

Our series confirms the capacity of the MEMMAT regimen to control relapsing/refractory pediatric brain malignancies beyond medulloblastomas. It seems that it would lead to better outcome if used as a second-line treatment and as part of maintenance. Performing additional state-of-the-art studies is mandatory to confirm or question our results.

## Figures and Tables

**Figure 1 jcm-12-01415-f001:**
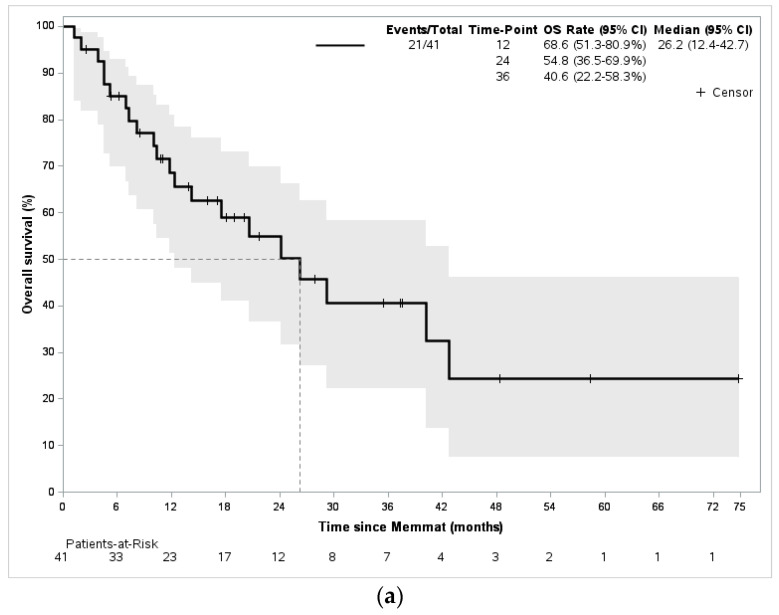
Overall survival and event-free survival of the whole population. (**a**) Overall survival; (**b**) event-free survival.

**Figure 2 jcm-12-01415-f002:**
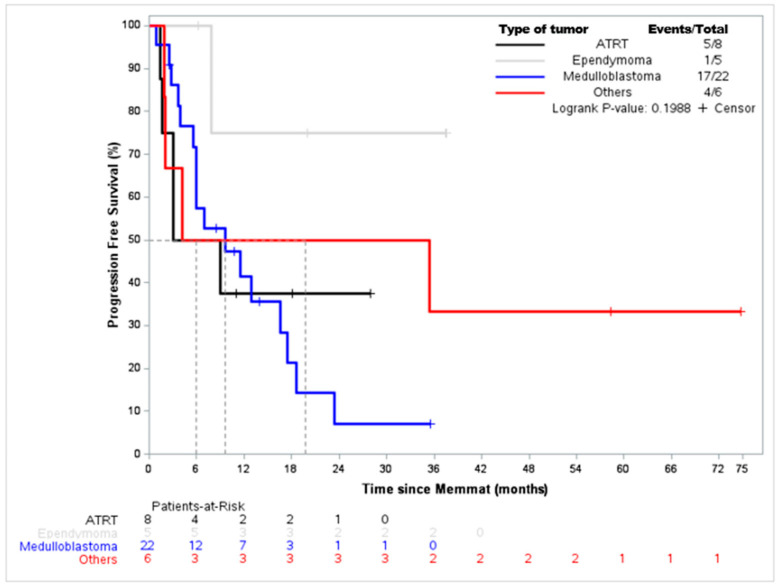
Event-free survival according to tumor type.

**Figure 3 jcm-12-01415-f003:**
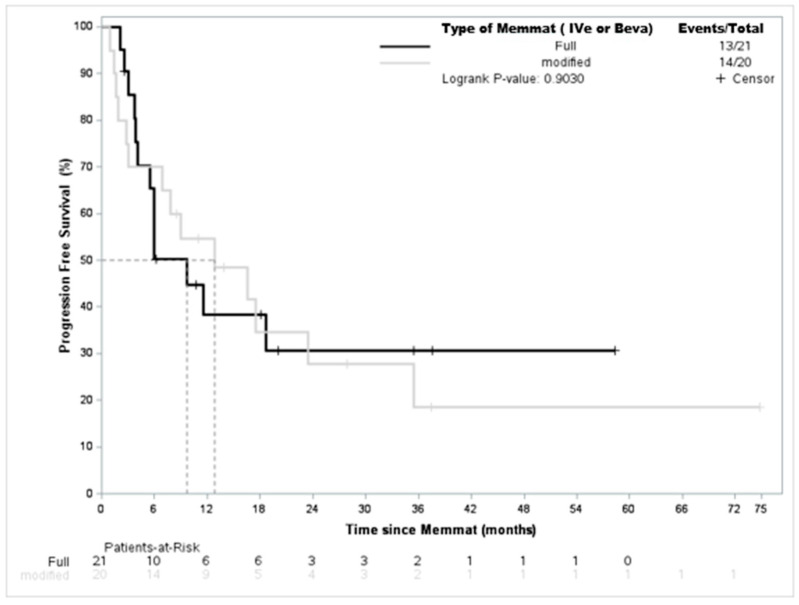
EFS according to treatment with full MEMMAT or modified MEMMAT.

**Figure 4 jcm-12-01415-f004:**
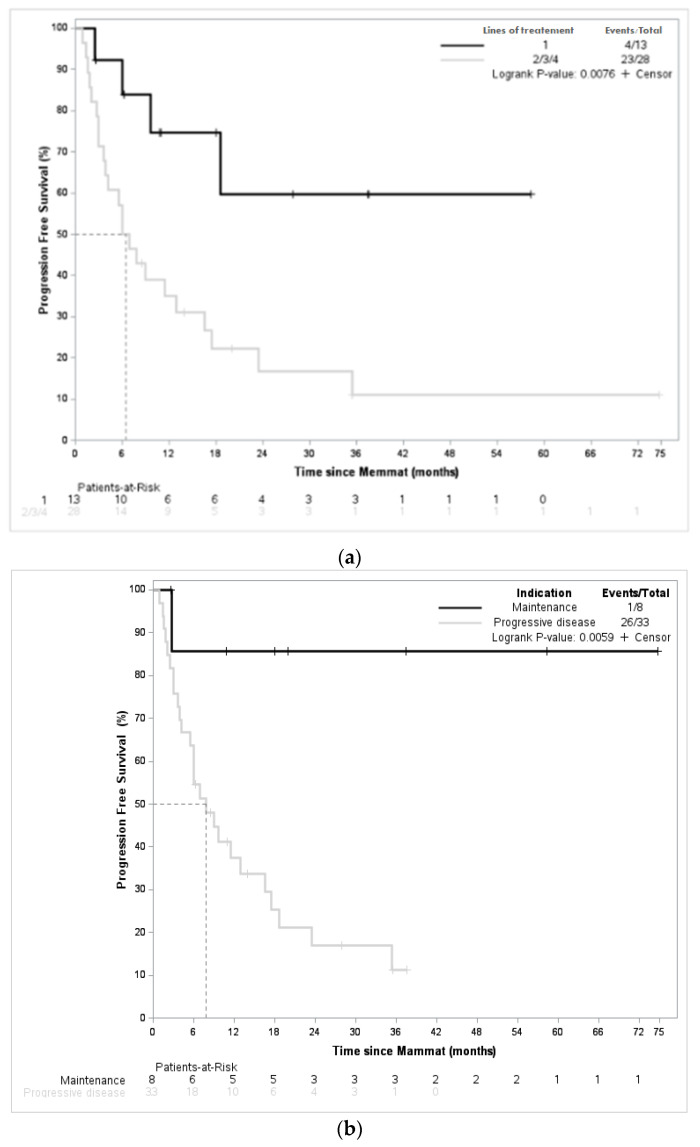
Event-free survival according to the number of previous lines of treatment and setting (progressive disease vs. maintenance). (**a**) EFS according to number of previous lines of treatment (1 vs. >1). (**b**) EFS according to the setting at the initiation of MEMMAT (maintenance vs. relapse).

**Table 1 jcm-12-01415-t001:** Patients characteristics.

Patient Characteristics	Frequency	
	N = 41	
**Age (years)**		
Median age (min; max)	14 (1; 19)	
**Gender**		
Female	20	(48.8%)
Male	21	(51.2%)
**Type of malignancy**		
Medulloblastoma	22	(53.7%)
Atypical Teratoid Rhabdoid Tumor	8	(19.5%)
Ependymoma	5	(12.2%)
Others	6	(14.6%)
**Number of previous lines of treatment**		
1	13	(31.7%)
2	17	(41.5%)
3	5	(12.2%)
4	6	(14.6%)
**Type of treatment previously received**		
Chemotherapy	39	(95.1%)
Surgery Radiotherapy Immunotherapy Targeted therapies	40 39 0 3	(97.6%) (95.1%) (0%) (7.3%)
**MEMMAT received**		
Full-type	16	(39.0%)
Modified-type	25	(61.0%)
**Drugs not administrated in modified MEMMAT**		
Bevacizumab	10	(24.4%)
Intrathecal injections Thalidomide	14 13	(34.1%) (31.7%)

**Table 2 jcm-12-01415-t002:** Observed grade III and IV toxicities.

Toxicity (Grade 3–4)	Number of Patients N = 41
**Hematological toxicity**	
-Neutropenia	20 (49%)
-Thrombopenia -Anemia	4 (10%) 2 (5%)
**Neurological disorder**	
-Neuropathy	7 (17%)
-Status epilepticus	2 (4%)
-Hemiparesis	1 (2%)
-Cerebellitis	1 (2%)
**Hemorrhagic disorder**	
-Macroscopic hematuria	1 (2%)
-Rectal bleeding	1 (2%)
**Intraventricular reservoir disorder**	3 (7%)
**Infections**	2 (5%)
**Others**	
-Mucositis -Rhinitis	2 (5%) 2 (5%)
-Thyroiditis	1 (2%)

**Table 3 jcm-12-01415-t003:** Characteristics of patients with ependymomas.

	Diagnosis	Tumor Status Prior to MEMMAT	nb Previous Lines of Treatment	Type of Treatment Previously Received	Time to Progression Prior to MEMMAT (days)	Time to Progression with MEMMAT (days)
Patient #1, 7 y-o	Anaplastic grade 3	local disease	1	Surgery, chemotherapy, radiation therapy	0	0
Patient #2, 21 y-o	Grade 2	local disease	3	Surgery, chemotherapy, radiation therapy	122	0
Patient #3, 1 y-o	Anaplastic grade 3	Metastatic disease, CSF positive	4	Surgery, chemotherapy, radiation therapy	240	34
Patient #4, 9 y-o	Anaplastic grade 3	Metastatic disease	1	Surgery, radiation therapy	452	0
Patient #5, 4 y-o	No data	Metastatic disease	1	Surgery, chemotherapy, radiation therapy	55	0

## Data Availability

The data will be available upon reasonable request to the authors.

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
