# Peer review of "Retrospective National “Real Life” Experience of the SFCE with the Metronomic MEMMAT and MEMMAT-like Protocol"

_jcm, 2023, doi:10.3390/jcm12041415_

Round 1

Reviewer 1 Report (Previous Reviewer 2)

The author has still not fully addressed the comments; for example, the MB sub-groups are well defined, but in the current study there is not much explanation, only a few sentences in the discussion, which is not helpful for the readers or the scientific community. The quality of the figures is poor.

Author Response

Reviewer 1:

  • The author has still not fully addressed the comments; for example, the MB sub-groups are well defined, but in the current study there is not much explanation, only a few sentences in the discussion, which is not helpful for the readers or the scientific community.

--> We are not quite sure we understand well this very comment. Is the reviewer suggesting a short section about the meaning of medulloblastoma grouping ?  We do not think it is necessary here. We mention a reference at the end of the manuscript that readers can refer to. Otherwise, we might also add a section on ATRT and Ependymoma subgrouping that also have prognostic and biological relevance. We do not thing this is within the scope of this manuscript.

Otherwise, as previously answered, the grouping analysis of the tumors is not available in this retrospective study. We have tried to discuss the fact the grouping for medulloblastoma does not seem to impact on the outcome when patients with medulloblastoma are treated with MEMMAT. We have added that group3 usually have the worse prognostic   than Group 4 medulloblastoma usually grow more slowly BUT but that Group 3 do better than group 4 when treated with MEMMAT to try to clarify things and turn this into practical information for the readers.

This section now goes like this:

Lastly, we do not have precise molecular subgrouping for most patients. Thus, for medulloblastoma for instance, it has been reported that Group 4 medulloblastoma had a less aggressive behavior (38). The repartition of the molecular subtypes of brain tumors included might therefore influence the EFS in this series. Anyhow, in the series from Slavc and al. (31), among the 23 patients whose tumors underwent molecular subgrouping classification using DNA methylation array and brain classifier mnp-V12.5, group 4 medulloblastomas seem to be doing poorer than group 3 medulloblastoma (median survival 43 months (14-71) vs not reached). As group 3 medulloblastoma are commonly accepted to display a worse prognostic, the findings reported by Slavc and al. (31) seem to further highlight the potential of MEMMAT for relapsing both Goup3 and 4 medulloblastomas which represent medical unmet needs.  Moreover, the lack of molecular profiling for most of our patient shall not minimize the interesting outcome reported despite they do not allow to move into the direction of precision medicine.  

  • The quality of the figures is poor.

--> We are sorry for this- we have provided HQ figure. I guess, the poor quality might be  the results of the transformation of the files during the processing of the manuscript. I guess we will let the editor tell us if the original figures are acceptable for the article when/if the manuscript is accepted for publication.

Reviewer 2 Report (Previous Reviewer 1)

Dear Authors,

please note that some errors reported in the last review  have not been corrected.:

- line 214: when you write: "among the patients who received MEMMAT as a manteinance 3 had medullo, 2 ependymoma and 1 ATRT" I am inclined to think that these patients are 6 but they are 8! please correct the sentence

- The point 4) of  my last review has not been corrected: I rewrite it because for me is very crucial. " I think that it is necessary to include a comment related to the fact that using a manteinance therapy might make it difficult to attribute treatment efficay to the manteinance only. It is more likely to pose to be the association with previous therapy, so it is important to specify this." Also in line 188 it could be useful introduce the concept that if  7 patients  have received radiotherapy it is not possible to understand with certainty the real role of MEMMAT.

- line 168: please correct with Table 2 (table 3 is related to ependymoma...)

There are a lot of typo errors :

- line 51 correct with BRAF pathway

- line 66: please insert LMIC abbreviation   which is written later in the text  (at line 289)

- please correct the use of brackets at line 153, in the caption of Table 2, line 199, 223, 303

- line 229: please insert "disease" after progressive

- fig 4 A please correct line instead of ligne

- line 272: please insert what MRD means

- please correct at line 300 the name  Slavc

- line 168 please correct double %%

Author Response

  • line 214: when you write: "among the patients who received MEMMAT as a manteinance 3 had medullo, 2 ependymoma and 1 ATRT" I am inclined to think that these patients are 6 but they are 8! please correct the sentence

--> This is indeed correct: 8 patients received maintenance. We gave precisions ly the 2 most frequent tumor types. We have now modified this sentence and added: and 3 patients with other type of tumors.

  • The point 4) of  my last review has not been corrected: I rewrite it because for me is very crucial. " I think that it is necessary to include a comment related to the fact that using a manteinance therapy might make it difficult to attribute treatment efficay to the manteinance only. It is more likely to pose to be the association with previous therapy, so it is important to specify this." Also in line 188 it could be useful introduce the concept that if  7 patients  have received radiotherapy it is not possible to understand with certainty the real role of MEMMAT.

--> We do agree with this very comment. There is currently no known salvage therapy for medulloblastoma if patients relapse after having received radiotherapy. Anyhow we do agree that when several different treatments are given, it is impossible to make definitive statement regarding the specific impact of MEMMAT in the outcome. We have introduced a sentence in the discussion about this issue (line 275):                     

Noteworthy in this perspective, survival is better when MEMMAT is given as a maintenance after additional treatment at relapse (i.e., surgery, re-irradiation, chemotherapy), although the several treatments make it impossible to formally demonstrate the respective impact of the different parts of the treatment on the outcome, the addition of MEMMAT as a maintenance seems to improve survival in our cohort.

  • line 168: please correct with Table 2 (table 3 is related to ependymoma...) :

--> this refers to Figure 2 with Kaplan Meier curves according to tumor types and not the Table 3 related to ependymoma.

  • There are a lot of typo errors:

-->  We thank the reviewer for his/her edits. Typos/errors have been corrected as followed. Of note the lines number do not appear to be the same on our version on the manuscript and the reviewers’…

- line 51 correct with BRAF pathway: this has been corrected

- please insert LMIC abbreviation which is written later in the text (at line 289) : this has been added

- please correct the use of brackets at line 153, in the caption of Table 2, line 199, 223, 303

- line 229: please insert "disease" after progressive: disease has been added

- fig 4 A please correct line instead of ligne.: this has been corrected in the figure

- line 272: please insert what MRD means: this has been clarified (minimal residual disease)

- please correct at line 300 the name  Slavc ; this has been corrected

- line 168 please correct double %% : This has been corrected

Round 2

Reviewer 1 Report (Previous Reviewer 2)

N/A

This manuscript is a resubmission of an earlier submission. The following is a list of the peer review reports and author responses from that submission.

Round 1

Reviewer 1 Report

Congratulations for the paper. It is interesting for the approach with a metronomic therapy in relapsed refractory setting in rare tumors that are almost all orphans of indications at relapse and with very poor prognosis.
The major comment is correlated to the necessity of better describe (in RESULTS and DISCUSSION)
patient characteristics, especially the ependymoma group of 5 patients (perhaps with a Table with previous treatments, type of treatments, times to relapse and times to the beginning of MEMMAT and  disease extension). In addition, it would be useful to better describe the 8 patients who performed MEMMAT as maintenance and their specific survival.

There are some evidences in literature that ependymomas could be treated at II/III lines with surgery and/or reirradiation with a significant improving of survival; so I think their survival in your cohort could be correlated with the extension of disease and the previous line of treatment (i.e. surgery? reirradiation?).

Could you specify the median follow-up?

There are some minor comments on:
1) English/typing  at lines:(38, 39, 49, 95, 98,99, 147, 162, 204, 279)

2) Table 1 and 3 are without any legend and without the correlated number and table 2 is missing and there are some typing errors in the two Tables
3) in Matherial and Methods is written that the entire cohort is for patients  youger than 19 years but in  Table of Patient Characteristics is written  a  range of 1-29 years. So I think something needs to be corrected.

4) Figure 1 (a) and (b) are not evident in the curves

5) The legend in Figure 2 is very busy and not so clear

6) At line 260 is written Kivivori et al (refereces 37), but it is the last name and in the previous line 259 is written Peyr et al  that is the first name of the references 26, so you should use the same criteria for all citations.

Author Response

JCM (ISSN 2077-0383): Retrospective national “real life” experience of the SFCE with the metronomic MEMMAT and MEMMAT-like protocol.

Point by point comments to the reveiwers.

Reviewer 1:

  • Congratulations for the paper. It is interesting for the approach with a metronomic therapy in relapsed refractory setting in rare tumors that are almost all orphans of indications at relapse and with very poor prognosis.

We thank the reviewer for this comment

  • Themajor comment is correlated to the necessity of better describe (in RESULTS and DISCUSSION) patient characteristics, especially the ependymoma group of 5 patients (perhaps with a Table with previous treatments, type of treatments, times to relapse and times to the beginning of MEMMAT and disease extension). In addition, it would be useful to better describe the 8 patients who performed MEMMAT as maintenance and their specific survival.

we do agree with the comment. We have added a table to better describe the ependymoma cohort that includes the parameter suggested by the reviewer a small section in the result section. Similarly, more details about the patients who received maintenance has been in the dedicated section of the results.

  • There are some evidences in literature that ependymomas could be treated at II/III lines with surgery and/or reirradiation with a significant improving of survival; so I think their survival in your cohort could be correlated with the extension of disease and the previous line of treatment (i.e. surgery? reirradiation?).

we do agree that surgery + re-irradiation + surgery at relapse can lead to significant prolonged survival in patients with medulloblastoma. Anyhow, only two patients in our series received MEMMAT as maintenance and one had local disease. 3 patients had metastatic disease at relapse. It is impossible here to correlate survival with maintenance or metastatic status given the limited number of patients. We have nevertheless added in the sentence in the dicussion.

  • Could you specify the median follow-up?

The median follow-up is 28 months. This has been added ion the results section.

There are some minor comments on:

  • English/typing  at lines:(38, 39, 49, 95, 98,99, 147, 162, 204, 279)

The typing errors have been corrected.

  • Table 1 and 3 are without any legend and without the correlated number and table 2 is missing and there are some typing errors in the two Tables

 the title of table been added and reordered with one table added following the reveiwer’s suggestion (see comment 2)

  • in Matherial and Methods is written that the entire cohort is for patients youger than 19 years but in  Table of Patient Characteristics is written  a  range of 1-29 years. So I think something needs to be corrected.

we thank the reviewer for noticing this typo. This has been corrected.

  • Figure 1 (a) and (b) are not evident in the curves.The legend in Figure 2 is very busy and not so clear

this was related to the small size of the figure. We have now enlarged the figures so that they are easy to read.

  • 6) At line 260 is written Kivivori et al (refereces 37), but it is the last name and in the previous line 259 is written Peyr et al  that is the first name of the references 26, so you should use the same criteria for all citations.

we thank the reviewer. The first name of the first has been corrected in the text ( we wrote the name of the last author instead.

Reviewer 2

  • The manuscript by Camille et al., showed the "real-life" treatment in pediatric brain tumors. The result is not as consistent with others but may offer another option. T

we do not agree with this very comment. Our results seems quite consistent with those reported previously in meeting  by the Vienna team and or by the Boston team with the 4 and 5 drug similar regimen for ependymoma cohorts.

  • All the patients without any markers (SHH, WNT..) to guide the treatment make this result weaker.

we do agree that the lack of biomarker to establish the molecular subtypes especially of medulloblastoma cohort weaken the results. Anyhow, quite counterintuitively group 4 seem to be bdoing not as good as group 3 as reported recently by the Vienna team when using the MEMMAT protocole. Lastly, all the medulloblastoma had previously recevied radiotherapy so that even SHH, WNT medulloblastoma would have a poor outcome. We have added a section to discuss the lack of molecular subtyping in our cohort.

  • And it will be important to analyze the molecular mechanism (RNAseq) before and after treatment. 

We do agree it would be both interesting and important, but this is a very different work than the reporting the clinical activity in a real life setting.

Of note, we have update the ref31 and changed the abstract from a meeting for the recently paper published in Cancers.

Reviewer 2 Report

The manuscript by Camille et al., showed the "real-life" treatment in pediatric brain tumors. The result is not as consistent with others but may offer another option. There are major concerns as follows:

1. All the patients without any markers (SHH, WNT..) to guide the treatment make this result weaker.

2. And it will be important to analyze the molecular mechanism (RNAseq) before and after treatment. 

Author Response

JCM (ISSN 2077-0383): Retrospective national “real life” experience of the SFCE with the metronomic MEMMAT and MEMMAT-like protocol.

Point by point comments to the reveiwers.

Reviewer 1:

  • Congratulations for the paper. It is interesting for the approach with a metronomic therapy in relapsed refractory setting in rare tumors that are almost all orphans of indications at relapse and with very poor prognosis.

We thank the reviewer for this comment

  • The major comment is correlated to the necessity of better describe (in RESULTS and DISCUSSION) patient characteristics, especially the ependymoma group of 5 patients (perhaps with a Table with previous treatments, type of treatments, times to relapse and times to the beginning of MEMMAT and disease extension). In addition, it would be useful to better describe the 8 patients who performed MEMMAT as maintenance and their specific survival.

we do agree with the comment. We have added a table to better describe the ependymoma cohort that includes the parameter suggested by the reviewer a small section in the result section. Similarly, more details about the patients who received maintenance has been in the dedicated section of the results.

  • There are some evidences in literature that ependymomas could be treated at II/III lines with surgery and/or reirradiation with a significant improving of survival; so I think their survival in your cohort could be correlated with the extension of disease and the previous line of treatment (i.e. surgery? reirradiation?).

we do agree that surgery + re-irradiation + surgery at relapse can lead to significant prolonged survival in patients with medulloblastoma. Anyhow, only two patients in our series received MEMMAT as maintenance and one had local disease. 3 patients had metastatic disease at relapse. It is impossible here to correlate survival with maintenance or metastatic status given the limited number of patients. We have nevertheless added in the sentence in the dicussion.

  • Could you specify the median follow-up?

The median follow-up is 28 months. This has been added ion the results section.

There are some minor comments on:

  • English/typing  at lines:(38, 39, 49, 95, 98,99, 147, 162, 204, 279)

The typing errors have been corrected.

  • Table 1 and 3 are without any legend and without the correlated number and table 2 is missing and there are some typing errors in the two Tables

 the title of table been added and reordered with one table added following the reveiwer’s suggestion (see comment 2)

  • in Matherial and Methods is written that the entire cohort is for patients youger than 19 years but in  Table of Patient Characteristics is written  a  range of 1-29 years. So I think something needs to be corrected.

we thank the reviewer for noticing this typo. This has been corrected.

  • Figure 1 (a) and (b) are not evident in the curves.The legend in Figure 2 is very busy and not so clear

this was related to the small size of the figure. We have now enlarged the figures so that they are easy to read.

  • 6) At line 260 is written Kivivori et al (refereces 37), but it is the last name and in the previous line 259 is written Peyr et al  that is the first name of the references 26, so you should use the same criteria for all citations.

we thank the reviewer. The first name of the first has been corrected in the text ( we wrote the name of the last author instead.

Reviewer 2

  • The manuscript by Camille et al., showed the "real-life" treatment in pediatric brain tumors. The result is not as consistent with others but may offer another option. T

we do not agree with this very comment. Our results seems quite consistent with those reported previously in meeting  by the Vienna team and or by the Boston team with the 4 and 5 drug similar regimen for ependymoma cohorts.

  • All the patients without any markers (SHH, WNT..) to guide the treatment make this result weaker.

we do agree that the lack of biomarker to establish the molecular subtypes especially of medulloblastoma cohort weaken the results. Anyhow, quite counterintuitively group 4 seem to be bdoing not as good as group 3 as reported recently by the Vienna team when using the MEMMAT protocole. Lastly, all the medulloblastoma had previously recevied radiotherapy so that even SHH, WNT medulloblastoma would have a poor outcome. We have added a section to discuss the lack of molecular subtyping in our cohort.

  • And it will be important to analyze the molecular mechanism (RNAseq) before and after treatment. 

We do agree it would be both interesting and important, but this is a very different work than the reporting the clinical activity in a real life setting.

Of note, we have update the ref31 and changed the abstract from a meeting for the recently paper published in Cancers.

Round 2

Reviewer 1 Report

-Please see some typo errors at line 39 (3/4), line 165, 222,all the paper need to be corrected better

- At line 228  after 'progressive' you can insert 'disease' or only 'after progression'

- at line 213-216 it is not clear  hiow many patients received MEMMAT as manteinance (9?): 3 with medullo, 2 with ependymoma, and the others?, in the curve at figure 4 B in the manteinance there are 8 patients, please correct and given that manteinace cohort seems to be important in this paper it woulb be useful to describe all the cohort.

- I think that it is necessary to include a comment related to the fact that using a manteinance therapy might make it difficult to attribute treatment efficay to the manteinance only. It is more likely to pose to be the association with previous therapy, so it is important to specify this.

Author Response

Retrospective National “Real Life” Experience of the SFCE with the Metronomic MEMMAT and MEMMAT-like Protocol _ jcm-1980054 R2

We thank the reviewer for his additional valuable comments.

Please find bellow our pout by point response.

1) Please see some typo errors at line 39 (3/4), line 165, 222,all the paper need to be corrected better

2)  At line 228  after 'progressive' you can insert 'disease' or only 'after progression'

--> the typos noted by the reviewer have been corrected. A careful editing of the manuscript has also been performed.

3)  at line 213-216 it is not clear  hiow many patients received MEMMAT as manteinance (9?): 3 with medullo, 2 with ependymoma, and the others?, in the curve at figure 4 B in the manteinance there are 8 patients, please correct and given that manteinace cohort seems to be important in this paper it woulb be useful to describe all the cohort.

--> We do agree it was important to have the maintenance data described here as previously mentioned after the first round of review. The number of patient has been corrected. It is indeed 8. We have added that among those patients 1 also had ATRT.

4) I think that it is necessary to include a comment related to the fact that using a manteinance therapy might make it difficult to attribute treatment efficay to the manteinance only. It is more likely to pose to be the association with previous therapy, so it is important to specify this.

--> we do agree with this comment and added an additional sentence in the ependymoma/maintenance section of the discussion.